# In Vitro Functionality and Endurance of GMP-Compliant Point-of-Care BCMA.CAR-T Cells at Different Timepoints of Cryopreservation

**DOI:** 10.3390/ijms25031394

**Published:** 2024-01-23

**Authors:** Genqiao Jiang, Brigitte Neuber, Angela Hückelhoven-Krauss, Uta E. Höpken, Yuntian Ding, David Sedloev, Lei Wang, Avinoam Reichman, Franziska Eberhardt, Martin Wermke, Armin Rehm, Carsten Müller-Tidow, Anita Schmitt, Michael Schmitt

**Affiliations:** 1Department of Internal Medicine V, University Clinic Heidelberg, 69120 Heidelberg, Germany; genqiao.jiang@stud.uni-heidelberg.de (G.J.); brigitte.neuber@med.uni-heidelberg.de (B.N.); angela.hueckelhoven-krauss@med.uni-heidelberg.de (A.H.-K.); yuntian.ding23@gmail.com (Y.D.); david.sedloev@med.uni-heidelberg.de (D.S.); xjwl8587@gmail.com (L.W.); avinoam.reichman@googlemail.com (A.R.); franziska.eberhardt@stud.uni-heidelberg.de (F.E.); carsten.mueller-tidow@med.uni-heidelberg.de (C.M.-T.); anita.schmitt@med.uni-heidelberg.de (A.S.); 2Department of Translational Tumor Immunology, Max-Delbrück Center for Molecular Medicine (MDC), 13125 Berlin, Germany; uhoepken@mdc-berlin.de (U.E.H.); arehm@mdc-berlin.de (A.R.); 3Early Clinical Trial Unit (ECTU), Medical Clinic and Poliklinik I, Carl Gustav Carus University, 01307 Dresden, Germany; martin.wermke@uniklinikum-dresden.de

**Keywords:** multiple myeloma, CAR-T cells, B-cell maturation antigen (BCMA), timepoint, stability and function of CAR-T cells

## Abstract

The search for target antigens for CAR-T cell therapy against multiple myeloma defined the B-cell maturation antigen (BCMA) as an interesting candidate. Several studies with BCMA-directed CAR-T cell therapy showed promising results. Second-generation point-of-care BCMA.CAR-T cells were manufactured to be of a GMP (good manufacturing practice) standard using the CliniMACS Prodigy^®^ device. Cytokine release in BCMA.CAR-T cells after stimulation with BCMA positive versus negative myeloma cell lines, U266/HL60, was assessed via intracellular staining and flow cytometry. The short-term cytotoxic potency of CAR-T cells was evaluated by chromium-51 release, while the long-term potency used co-culture (3 days/round) at effector/target cell ratios of 1:1 and 1:4. To evaluate the activation and exhaustion of CAR-T cells, exhaustion markers were assessed via flow cytometry. Stability was tested through a comparison of these evaluations at different timepoints: d0 as well as d + 14, d + 90 and d + 365 of cryopreservation. As results, (1) Killing efficiency of U266 cells correlated with the dose of CAR-T cells in a classical 4 h chromium-release assay. There was no significant difference after cryopreservation on different timepoints. (2) In terms of endurance of BCMA.CAR-T cell function, BCMA.CAR-T cells kept their ability to kill all tumor cells over six rounds of co-culture. (3) BCMA.CAR-T cells released high amounts of cytokines upon stimulation with tumor cells. There was no significant difference in cytokine release after cryopreservation. According to the results, BCMA.CAR-T cells manufactured under GMP conditions exerted robust and specific killing of target tumor cells with a high release of cytokines. Even after 1 year of cryopreservation, cytotoxic functions were maintained at the same level. This gives clinicians sufficient time to adjust the timepoint of BCMA.CAR-T cell application to the patient’s course of the underlying disease.

## 1. Introduction

Multiple myeloma (MM), the second most common hematologic disease, is characterized by the uncontrolled proliferation of plasma cells [1,2]. Due to the development of several new therapeutic agents against MM over the last two decades, overall survival rates have significantly increased [3]. After the revolution of allogeneic stem cell therapy, immunomodulatory imide drugs (IMiDs) such as thalidomide, lenalidomide and monoclonal antibodies such as daratumumab and isatuximab, directed against CD38, elotuzumab directed against SLAMF-7, and conjugate belantamab–mafodotin have all contributed to an increased survival rate [4]. Despite the availability of these groundbreaking treatment options, MM remains generally incurable and it claimed 117,077 lives worldwide in 2020 [5]. Patients who are refractory to these now-standard treatments still have an especially poor prognosis, often surviving less than a year after diagnosis [6,7,8]. This group of patients has an extremely urgent need for new therapeutic options.

Chimeric antigen receptor (CAR)-T cell (CAR-T) therapy has become a novel therapeutic option for cancer patients, most effectively in CD19.CAR-T cells in patients with B-cell lymphomas or B-cell acute lymphoblastic leukemia. CAR-T cells are manufactured to bind the antigen-binding sites of antibodies and T cells can be activated against the cancer antigen. CAR-T cell therapy also opens new avenues in treatment of MM patients.

Different target antigens for CAR-T cell therapy against MM cells were studied and several molecules (CD138, CD38, SLAMF7, NY-ESO-1) were identified as interesting targets [9]. To decrease severe side effects, antigens should be highly expressed in the tumor tissue, with low expression in peripheral tissue [10]. B-cell maturation antigen (BCMA), also referred to as TNFRSF17 or CD269, a 184 amino-acid-long transmembrane glycoprotein, and a member of the tumor necrosis factor superfamily, fulfills this criterion. It is expressed in mature B-cells and plays an inherent part in the proliferation and survival of plasma cells. However, BCMA is expressed at a very low level in other healthy tissues simultaneously [11,12]. Several studies of BCMA-directed CAR-T cell therapy showed highly promising results leading to the FDA and EMA approval of the first BCMA-directed CAR construct in 2021. Abecma™ (ide-cel) is approved for patients with refractory disease after at least three prior lines of treatment [13,14,15], while Carvykti^TM^ (cilta-cel) led to substantial and durable responses and remained favorable with longer follow-up [16,17].

Besides commercial BCMA.CAR-T cell products, point-of-care CAR-T cell products are also under current clinical investigation. However, clinical responses last only 8–11 months [13,18]. In our clinical trial CARLOTTA (CT-2022-502831-20-00-IN-003), we will try a novel BCMA.CAR with high affinity suggesting a long-lasting clinical response. As a newly promoted immune therapy, BCMA.CAR-T cell therapy requires appropriate methods to maintain a stable efficacy to kill tumor cells after storage or long-distance transportation. Long- or short-time cryopreservation, therefore, needs careful assessment. This was performed for point-of-care CAR-T cell products in this present work.

## 2. Results

### 2.1. Cytokine Release Assays

To assess the functionality of BCMA.CAR-T cells, cytokine release was evaluated at four different timepoints (fresh, 14 days, 90 days and 1 year from cryopreservation) from three different healthy donors. Values for positive cytokine release were defined as percentage of cytokine-producing BCMA.CAR-T cells for TNFα and/or IFNγ, while the transduction efficiency of BCMA.CAR-T cells is shown in Table 1. This was evaluated via intracellular staining after stimulation with BCMA-positive/negative tumor cells. Figure 1 and Figure 2 show the results of cytokine-producing cells.

For each donor, there is no significant difference between the group without stimulation and the group with HL60 stimulation [(0.97 ± 0.64)% and (1.81 ± 0.71)%, *p* = 0.121; (0.85 ± 0.51)% and (1.30 ± 0.69)%, *p* = 0.295; (1.25 ± 0.22)% and (2.54 ± 0.95)%, *p* = 0.051], but there is a significant difference between the group without stimulation and the group with U266 stimulation [(0.97 ± 0.64)% and (21.54 ± 4.70)%, *p* = 0.003; (0.85 ± 0.51)% and (19.51 ± 4.38)%, *p* = 0.003; (1.25 ± 0.22)% and (62.31 ± 1.67)%, *p* < 0.0001]. (Figure 1A,B).

To evaluate and differentiate the functionality of BCMA.CAR-T cells in vitro before and after cryopreservation, we compared cytokine-producing cells at different timepoints. There was no significant difference after cryopreservation (d + 14, d + 90 and d + 365): neither double-positive (TNFα- and IFNγ-producing) BCMA.CAR-T Cells (*F* = 3.39, *p* = 0.173), nor single-positive (TNFα- or IFNγ-producing) BCMA.CAR-T cells (*F* = 1.72, *p* = 0.237; *F* = 0.04, *p* = 0.850) (Figure 1C).

For BCMA.CAR-T cells, the releases of TNFα and IFNγ were significantly different in each donor [(59.41 ± 7.36)% and (20.84 ± 5.38)%, *p* = 0.003; (66.10 ± 13.87)% and (19.51 ± 4.84)%, *p* = 0.013; (77.82 ± 9.06)% and (62.17 ± 0.95)%, *p* = 0.044], which is shown in Figure 2A. In CD8^+^ BCMA.CAR-T cells, the cells producing TNFα and IFNγ had comparable levels [(49.26 ± 8.86)% and (35.25 ± 7.50)%, *p* = 0.052; (46.36 ± 13.71)% and (34.73 ± 7.33)%, *p* = 0.112; (75.64 ± 6.51)% and (70.77 ± 5.72)%, *p* = 0.252]. However, in CD4^+^ BCMA.CAR-T cells, the value for IFNγ was significantly lower than for TNFα [(14.35 ± 8.37)% and (61.76 ± 8.64)%, *p* = 0.003; (14.10 ± 5.01)% and (69.86 ± 13.03)%, *p* = 0.004; (52.15 ± 6.04)% and (80.87 ± 9.15)%, *p* = 0.006]. These different levels of cytokine-producing cells in CD4^+^ versus CD8^+^ BCMA.CAR-T cells and in all CAR-T cells are also shown in Figure 2B,C.

### 2.2. Cytotoxic Assays

For an evaluation of the cytotoxic potency of BCMA.CAR-T cells, an in vitro ^51^Cr release assay was used. We found that the killing of tumor cells correlated with the dose of CAR-T cells, as shown in Figure 3. Furthermore, only BCMA-expressing myeloma cells, i.e., U266, were killed. There was no killing effect for BCMA-negative tumor cells, i.e., HL60. Killing potency was seen at different timepoints.

To determine the stability of BCMA.CAR-T cell function, especially before and after cryopreservation, we compared all results at different timepoints. For killing efficiency, i.e., short-term cytotoxic potency (^51^Cr) after cryopreservation (14 d, 90 d or 1 y), there was no significant difference (*F* = 3.385, *p* = 0.125), either. 

### 2.3. Co-Culture Assays

The killing status of BCMA.CAR-T cells was assessed using co-culture assays and flow cytometry afterwards. After three rounds (at day 10) of co-culture, BCMA.CAR-T cells reached the dominant position while U266 tumor cells nearly disappeared. In the control group, with the incubation of non-transduced T cells, U266 tumor cells showed overgrowth while non-transduced T cells did not proliferate. Figure 4 shows the percentage of different cells in different co-culture groups. 

To evaluate the endurance of BCMA.CAR-T cell function in vitro, we performed the co-culture assay at timepoints of 90 days and 1 year after cryopreservation. Over around six rounds, BCMA.CAR-T cells kept their ability to eliminate all tumor cells as shown in Figure 5. At a 1:1 ratio, BCMA.CAR-T cells showed good killing efficiency and mostly maintained proliferation during all rounds. BCMA.CAR-T cells produced by the third donor stopped proliferation at the fourth round after 90 days of freezing, while after one-year of freezing, BCMA.CAR-T cells from the first donor experienced a proliferation collapse after four rounds. But the tumor cells were still restrained. At an E/T ratio of 1:4, donors showed individual phenomena: BCMA.CAR-T cells originating from donor 1 lost control of tumor cells two rounds earlier when they were frozen longer; for donor 2, the damping of BCMA.CAR-T cells happened at the fourth or fifth round, while tumor cells were revived to increase; BCMA.CAR-T cells from donor 3 were defeated even earlier at the third round at both timepoints.

### 2.4. Memory T Cells

After an evaluation of different memory T cell subsets, we found CAR-T_EM_ cells to be present always at a high level and to increase over time. CAR-T_EMRA_ cells were the second highest T cell subset, but, in contrast to CAR-T_EM_ cells, the frequency of CAR-T_EMRA_ cells declined over time. Naïve cells were infrequent in co-cultures and subsequently declined. T_CM_ cells remained at a low level and generally declined. In different memory T cell sets (T_EMRA_, T_EM_, T_N_ and T_CM_ cells), there was no significant difference between these two different timepoints at a 1:1 ratio [(19.10 ± 3.60)% and (29.94 ± 6.56)%, *p* = 0.063; (73.77 ± 5.64)% and (62.95 ± 11.75)%, *p* = 0.087; (2.58 ± 1.13)% and (1.70 ± 2.20)%, *p* = 0.318; (4.46 ± 0.81)% and (2.92 ± 3.07)%, *p* = 0.361] and 1:4 ratio [(32.57 ± 9.67)% and (32.03 ± 7.93)%, *p* = 0.929; (58.71 ± 12.09)% and (63.50 ± 9.80)%, *p* = 0.464; (2.31 ± 1.77)% and (1.61 ± 1.90)%, *p* = 0.053; (4.67 ± 1.52)% and (2.93 ± 2.74)%, *p* = 0.159], frozen after 90 days or 1 year. The frequency of different T cell subsets over time is shown in Figure 6.

### 2.5. Exhaustion of T Cells

Some surface markers on the T cells were detected to evaluate the exhaustion of BCMA.CAR-T cells: PD1 declined when CAR-T cells were activated and proliferated but increased when they failed to kill; LAG3 showed a similar trend as PD1 but TIM3 had a flat trend. No matter how high the ratio, the markers (PD1, LAG3 and TIM3) showed no significant difference between BCMA.CAR-T cells frozen for 90 days and 1 year (1:1 ratio: *p* = 0.184, *p* = 0.158, *p* = 0.065; 1:4 ratio: *p* = 0.413, *p* = 0.664, *p* = 0.110). The trends in different exhaustion markers are shown in Figure 7.

## 3. Discussion

The manufacturing of CAR-T cells of GMP quality requires many labor- and cost-intensive processes. To ensure the highest quality of this living drug and to warrant a safe and efficient therapy for the patients, each step of production and cryopreservation needs to be precisely validated. CAR-T cells have to be stored and transported before patients can receive the drug. Therefore, cryopreservation is an inherent part of the manufacturing process. Its influence on the enduring killing ability of CAR-T cells needs to be carefully characterized, especially as longer cryopreservation might improve and ensure supply chains for patients in need of CAR-T cell therapy. Lastly, it could reduce costs and therefore enable a wider use of this life-saving therapy. Our research shows that (1) frozen BCMA.CAR-T cells were most probably functional and released similar levels of TNFα and IFNγ after thawing and resting when compared with fresh CAR-T cells and that (2) the killing capacity of frozen BCMA.CAR-T cells was comparable to the capacity of fresh CAR-T cells [19].

In this study, BCMA.CAR-T cells manufactured according to GMP standards were derived from three healthy donors. To assess the durability and killing functionality of these cells, we employed three different assays. Each of the assays was performed at four different timepoints: fresh cells, and cryopreservation after 14 days, 90 days and 1 year. Previous work from our group suggested that the side effects of cryopreservation on CAR-T cells are transient and cells need time to recover from the harsh freeze–thaw process under certain conditions [20]. Therefore, we used the resting procedure after thawing the BCMA.CAR-T cells to preserve their function as much as possible.

Intracellular staining is widely used to detect cytokines released by cells after stimulation as a sign of cell activation. Therefore, we used tumor cell lines expressing BCMA, thus stimulating the BCMA.CAR-T cells. Obviously, higher levels of cytokines, like TNFα and IFNγ, were released by these stimulated BCMA.CAR-T cells. We discovered no reduction in the activation levels of TNFα and IFNγ even after cryopreservation for 14 days, 90 days or 1 year. This demonstrates that the capability of BCMA.CAR-T cells to produce cytokines did not decline over time. Some previous studies reported similar results for CD19.CAR-T cells [21,22]. The data also showed a constant trend toward a lower frequency of INFγ-producing cells than TNFα-producing cells over the entire time of cryopreservation of up to one year in all CAR-T cells. Interestingly, IFNγ release was more pronounced in CD8^+^ than in CD4^+^ CAR-T cells. Since more CAR-T cells are CD8^+^ than CD4^+^, CD8^+^ CAR-T cells exhibit a high capability for IFNγ production [23]. This might explain, at least partially, the difference in IFNγ release observed in the third donor when compared with the other two donors, as the CD4^+^/CD8^+^ T cell ratio in this third donor was substantially lower than in the other donors.

Previous reports suggested that the infusion of higher doses of CAR-T cells was associated with a better prognosis for patients with refractory B-cell malignancies [20,24]. To evaluate the short-term cytotoxicity of BCMA.CAR-T cells at different doses, we used a classical ^51^chromium release assay. We observed a dose-dependent specific lysis of tumor cells by BCMA.CAR-T cells from 30-fold to 1-fold. According to further data of frozen BCMA.CAR-T cells, the viability of cryopreserved BCMA.CAR-T cells remained constant over one year of cryopreservation.

To assess long-term killing efficiency, our three-round (10-day) co-culture demonstrated an excellent killing of BCMA.CAR-T cells. BCMA.CAR-T cells reached a dominant position, while tumor cells nearly disappeared in each round. Non-transduced T cells hardly had killing ability. Similar experiments were performed with BCMA.CAR-T cells after 14-day and 90-day cryopreservation. After thawing and resting, these frozen CAR-T cells maintained their cytotoxicity at a similarly high level. BCMA.CAR-T cells showed a long endurance of functionality after 90 days and 1 year of cryopreservation, as demonstrated by the results of a longer co-culture process. Equal amounts of effector cells and target cells were incubated and tumor cells were killed within six rounds of co-culture, while BCMA.CAR-T cells (at higher doses) kept proliferating. When BCMA.CAR-T cells were tested at lower doses with higher numbers of tumor cells, the failure of CAR-T cells occurred and tumor cell overgrowth was observed. However, this finding was donor-dependent.

Memory T cells were detected in co-culture assays. The results revealed that T_EM_ and T_EMRA_ were always dominant, even though their proportion changed with the proliferation of CAR-T cells. The stable killing efficiency during the co-culture process was demonstrated by the increasing proportion of differentiated effector cells. In contrast, T_N_ and T_CM_ cells were only detected at low frequencies. Furthermore, naïve cells vanished after 4 or 5 rounds of co-culture [25,26], or they might also have differentiated into other T cell subtypes. This non-significant difference confirmed that the distribution of different CAR-T cell subsets did not change after short- or long-time cryopreservation.

The exhaustion markers of BCMA.CAR-T cells were assessed during the co-culture process. We found that PD1 and LAG3, the immune-suppression checkpoints, declined until BCMA.CAR-T cells lost their function. When the trend of both markers changed, the killing function of CAR-T cells in the co-culture simultaneously also changed. Exhaustion markers were expressed on the surface of BCMA.CAR-T cells. Lower signals showed activation and a lack of exhaustion [27]. The marker TIM3 showed less significant changes and was kept at a stable and lower level, since TIM3 is a marker for the most dysfunctional T cell subset in cancer in line with other observations [28,29,30]. TIM3 showed co-expression with PD1 in both CD4^+^ and CD8^+^ T cells of preclinical cancer models [28,30]. 

## 4. Materials and Methods

BCMA.CAR-T cells were produced under GMP-grade conditions. We established a GMP core facility where we generated our own CAR-T products. Also, in our clinical department, BCMA.CAR-T cells were evaluated for functional efficiency, activity and stability at four different timepoints: fresh cells (after 12 days of production), as well as cells after 14 days, 90 days and 1 year of freezing. Cells were cryopreserved in liquid nitrogen (≤−140 °C) following a standard operating procedure (SOP). 

### 4.1. BCMA.CAR-T Cells Preparation

#### 4.1.1. Design and Produce

BCMA.CAR-T cells were manufactured in a fully closed system, the CliniMACS Prodigy System [27,31](Miltenyi Biotec, Bergisch Gladbach, Germany), which is an all-in-one CAR-T cell production platform located within the GMP facility of the University Hospital Heidelberg. Three healthy donors (V01–V03) were included in the study; all donors gave written informed consent, which was approved by the appropriate ethics committee. The GMP retroviral vector, pES.12-6(hBCMA-CAR) ps, was designed by MDC Berlin and manufactured by BioNTech. J22.9-FSY, a variant of a chimeric anti-human BCMA antibody (J22.9-xi), was selected for the construction of this fully humanized 2nd-generation CD28 CAR [32]. 

The complete procedure of the system for BCMA.CAR-T cell production is as follows: The starting materials, T cells, were obtained from the peripheral blood of healthy donors after leukapheresis. Before enrichment of CD8^+^ and CD4^+^ T cells, red blood cells (RBCs) were removed. Leukapheresis products were subjected to CD8 and CD4 immunomagnetic selection using CliniMACS^®^ CD8 and CliniMACS^®^ CD4 reagents (Miltenyi Biotec, Bergisch Gladbach, Germany). These T cells were activated by adding MACS^®^ GMP T Cell TransAct (Miltenyi Biotec, Bergisch Gladbach, Germany). After activation, retroviral transduction was performed using a SIN-vector. Transduced T cells were expanded in the presence of 12.5 ng/mL IL-7 and 12.5 ng/mL IL-15 cytokines until day 12. On day 12, the BCMA.CAR-T cells were harvested and cryopreserved. Quality controls (cell count and viability, transduction efficiency, vector copy number, mycoplasma and endotoxin level and sterility) were effectively performed during the manufacturing process and the final product was collected. The final products we used here were point-of-care CAR products.

#### 4.1.2. Transduction Efficiency

The expression of BCMA.CAR was detected using BCMA.CAR detection reagent (Miltenyi Biotec, Bergisch Gladbach, Germany) and Biotin Antibody, PE (Miltenyi Biotec, Bergisch Gladbach, Germany). The transduction efficiency was assessed via flow cytometry.

#### 4.1.3. Culture

BCMA.CAR-T cells and non-transduced T cells were cultured in Complete Medium (CM), which consists of 44.5% Click’s medium (FUJIFILM Irvine scientific, Tilburg, The Netherlands), 44.5% RPMI 1640 (Gibco^TM^, Carlsbad, CA, USA), 10% FBS (Gibco^TM^, Carlsbad, CA, USA) and 1% GlutaMax^TM^ supplement (Gibco^TM^, Carlsbad, CA, USA), at 37 °C with 5% CO_2_.

#### 4.1.4. Thaw and Rest

BCMA.CAR-T cells for stability research were stored in the liquid phase of liquid nitrogen tanks (≤−140 °C) until the timepoint of cell revival. They were thawed in a 37 °C water bath rapidly and washed twice with complete medium and cultured. Before each experiment, a standard resting procedure was performed for BCMA.CAR-T cells after washing [20]. Revived CAR-T cells were resuspended at ~2 × 10^6^ cells/mL concentration in CM, and 10 ng/mL IL-7 and 10 ng/mL IL-15 were added into the medium, additionally. Thereafter, CAR-T cells were cultured in a T25 flask and rested at normal culture condition for 18 h. 

### 4.2. Tumor Cell Lines

U266 (DSMZ ACC 003), a CD3^−^CD138^+^ human multiple myeloma cell line, and HL60 (DSMZ ACC 009), a CD3^−^CD138^−^ human acute myeloid leukemia cell line, authenticated by Leibniz Institute DSMZ-German Collection of Microorganisms and Cell Cultures GmbH, were cultured in RPMI 1640 medium, which contained 10% FBS and 2 mmol/L GlutaMax^TM^ supplement at 37 °C with 5% CO_2_. Cells were split 1:4 every 3 days.

### 4.3. Intracellular Staining (ICS) and Flow Cytometry Assays

Intracellular staining was used for detecting TNF-α- and IFN-γ-producing cells. After overnight resting, 2 × 10^5^ BCMA.CAR-T cells were incubated with 4 × 10^5^ U266 or HL60 cells for 5 h at 37 °C, 5% CO_2_. Brefeldin A (BFA, Biolegend, San Diego, CA, USA) and Monensin (MN, Biolegend, San Diego, CA, USA) were used for trapping cytokines. For the negative control, 2 × 10^5^ BCMA.CAR-T cells and non-transduced T cells were incubated in CM in the same conditions, while non-transduced T cells stimulated by 1 mg/mL Staphylococcal enterotoxin B from Staphylococcus aureus (SEB, Sigma-Aldrich, Burlington, MA, USA) were used as a positive control.

Dead cells were excluded using LIVE/DEAD Fixable Near-IR Dead Cell Stain Kit (Invitrogen, Carlsbad, CA, USA). After 5 h of stimulation, cells were harvested and washed in cold phosphate-buffered saline (PBS, Sigma-Aldrich, Burlington, MA, USA), followed by 30 min of NEAR-IR staining at 4 °C. Then, surface marker staining was performed with anti-CD3-Alexa Fluor 700 (Biolegend, San Diego, CA, USA), anti-CD4-BV510 (Biolegend, San Diego, CA, USA), anti-CD8-PerCP (Biolegend, San Diego, CA, USA) and anti-CD138-APC (Biolegend, San Diego, CA, USA) to distinguish different BCMA.CAR-T cell subsets and U266 cells. BCMA.CAR-T cells were stained using BCMA.CAR detection reagent (Miltenyi Biotec, Bergisch Gladbach, Germany), which can be readily identified using fluorochrome-labeled biotin antibodies, anti-biotin-PE (Miltenyi Biotec, Bergisch Gladbach, Germany). Afterwards, FoxP3 Staining Buffer Set (Miltenyi Biotec, Bergisch Gladbach, Germany) was used for fixation and permeabilization following the manufacturer’s protocol, and cytokine-producing cells were detected using anti-interferon (IFN)-γ-Dazzle 594 (Biolegend, San Diego, CA, USA) and anti-tumor necrosis factor (TNF)-α-BV421 (BD Biosciences, Franklin Lakes, NJ, USA). 

PBMCs from healthy donors were used for fluorescence compensation; they underwent the same process as the experimental groups and were applied before each data acquisition. Negative unstained controls were also used in the study. All cells requiring detection were harvested then washed twice with cold fluorescence-activated cell sorting (FACS) buffer (Miltenyi Biotec, Bergisch Gladbach, Germany). Then, 100 μL of cell suspension was transferred into a Falcon^®^ round-bottom polystyrene test tube (Corning, NY, USA) at a concentration of 1 × 10^7^ cells/mL and was stained with antibodies at the required conditions (followed the manufacturer’s protocols). All samples were analyzed via flow cytometry on the BD LSRII platform (BD Biosciences, Franklin Lakes, NJ, USA) and the data were analyzed using FlowJo (FlowJo, Ashland, OR, USA). Gating strategy was based on the recommendation from the International Multiconsortia Proficiency panel [33], as shown in Appendix A.

### 4.4. Functional Assays 

#### 4.4.1. Cytotoxicity Assay (Short-Term)

The cytotoxic efficiency of BCMA.CAR-T cells (effector cells, E) was assessed using a 4 h chromium-51 (^51^Cr; Hartmann Analytic, Braunschweig, Germany) release assay. The co-incubation ratios of effector cells and tumor cell lines, U266 or HL60 (target cells, T), were set at 30:1, 10:1, 3:1 and 1:1 (effector cells/target cells; E/T). Target cells were labeled with ^51^Cr for 2 h incubation before adding effector cells. Spontaneous release was controlled by incubation in CM alone and maximum release was detected by incubating target cells with Triton X-100 (Sigma-Aldrich, Burlington, MA, USA). After 4 h of incubation at 37 °C, 5% CO_2_, the supernatants were collected and the radioactivity was assessed via a 1414 WinSpectral^TM^ liquid scintillation counter (PerkinElmer, Waltham, MA, USA). For each ratio, triplicate experiments were performed. Specific lysis was calculated according to the following formula [34]:% Specific lysis = (^51^Cr release in the test well − spontaneous^51^Cr release)/(maximum ^51^Cr release − spontaneous ^51^Cr release) × 100.

#### 4.4.2. Co-Culture Assays (Long-Term) and Flow Cytometry Assays

BCMA.CAR-T cell cytotoxicity was assessed via a long-term co-culture of effector cells with target cells at E/T ratios of 1:1 or 1:4. Simultaneously, exhaustion markers and subtypes of the BCMA.CAR-T cells were measured. The first round of co-culture started with 15,000 effector cells and 15,000 or 60,000 target cells. Target cells were split every three days. Flow cytometric evaluation was repeated every third day (each round) and the assays for fresh and 14-day cryopreservation timepoints stopped at the third round. For 90-day and 1-year cryopreservation BCMA.CAR-T cells, flow cytometric evaluation was also repeated every three days (each round) but until six rounds of co-culture were completed. The aim of expansion is to investigate the durability of CAR-T cell killing efficiency.

The number of cells was evaluated using CountBright™ Absolute Counting Beads (Invitrogen, Carlsbad, CA, USA). BCMA.CAR-T cells were also stained using BCMA.CAR detection reagent and anti-biotin-PE, while U266 cells were stained using anti-CD138-APC/Cy7 (Biolegend, San Diego, CA, USA). Surface marker staining was performed using anti-CD3-Dazzle 594 (Biolegend, San Diego, CA, USA), anti-CD4-Alexa Fluor 700 (Biolegend, San Diego, CA) and anti CD8-PerCP (Biolegend, San Diego, CA, USA) to distinguish different CAR-T cell subsets. The following fluorochrome-conjugated antibodies anti-PD-1-Alexa-Fluor 488 (Biolegend, San Diego, CA, USA), anti-TIM-3-BV421 (Biolegend, San Diego, CA, USA) and anti-LAG-3-BV510 (Biolegend, San Diego, CA, USA) were used to detect the surface exhaustion markers of CAR-T cells. 

Surface markers CCR7 and CD45RA were stained using anti-CCR7-PE/Cy7 (Biolegend, San Diego, CA, USA) and anti-CD45RA-APC (Biolegend, San Diego, CA, USA) to assess subpopulation markers on memory T cell subsets as follows: naïve T cells (T_N_, CD45RA^+^ CCR7^+^), central memory T cells (T_CM_, CD45RA^−^ CCR7^+^), effector memory T cells (T_EM_, CD45RA^−^ CCR7^−^) and effector memory RA^+^ T cells (T_EMRA_, CD45RA^+^ CCR7^−^). PBMCs from healthy donors were used for fluorescence compensation; they were tested using the same process as the experimental groups and applied before each datum’s acquisition. Appropriate negative controls, including unstained control and fluorescence minus one (FMO) control, were also used in this study. All cells requiring surface marker detection were harvested and washed twice with cold FACS buffer, and then 100 μL of cell suspension was transferred into test tubes at a concentration of 1 × 10^7^ cells/mL. Cells were stained with the above antibodies at required conditions (followed manufacturer’s protocols). After the entire process of staining was completed, the counting beads were warmed to room temperature and vortexed for 30 s. Then, the beads were added into the cell suspension immediately. The microspheres are compatible with different excitation sources and emit fluorescence ranging from 385 nm to 860 nm; from this, the absolute number of cells was calculated. All samples were analyzed via flow cytometry on the BD LSRII platform and the data were analyzed using FlowJo 10.9.0.

### 4.5. Statistical Analysis

Statistical analysis was performed using Prism 9 (GraphPad Software, Inc., La Jolla, CA, USA). *p* values were calculated either using the two-way t test or one-way analysis of variance (ANOVA). *p* values < 0.05 were considered statistically significant. Graphs and tables were designed using Prism 9 (GraphPad Software, Inc., La Jolla, CA, USA). If not otherwise mentioned, the results are presented as mean ± SD.

## 5. Conclusions

BCMA.CAR-T cells manufactured under GMP conditions showed a robust and specific killing of target tumor cells with a high release of cytokines. Even after one year of cryopreservation, their cytotoxic function was maintained at the same level. This gives clinicians enough time to reschedule the timepoint of BCMA.CAR-T cell application based on the patient’s course of the underlying disease.

## Figures and Tables

**Figure 1 ijms-25-01394-f001:**
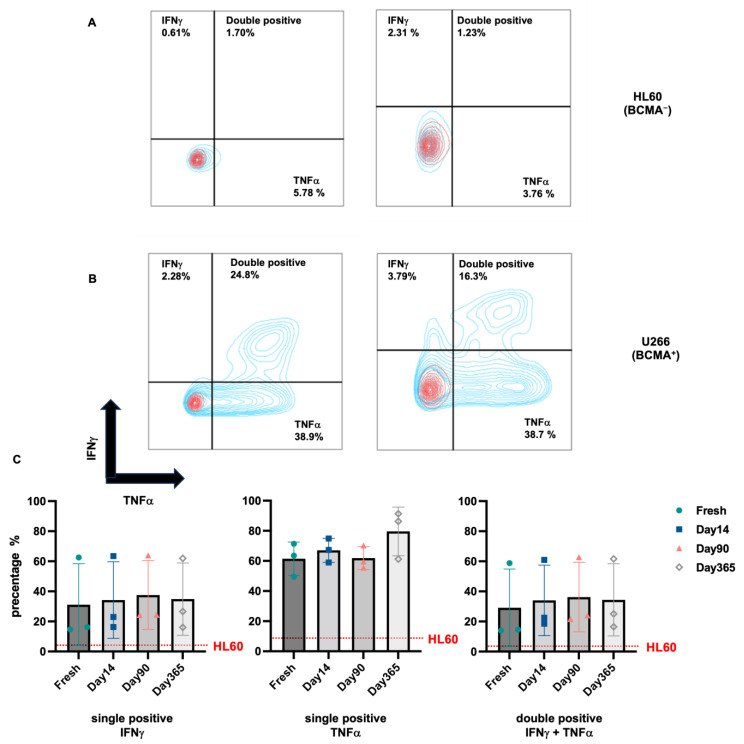
Percentage of BCMA.CAR-T cells with cytokine release of TNFα and IFNγ detected via intracellular staining. (**A**) shows stimulation of BCMA.CAR-T cells by the BCMA-negative tumor cell line HL60. CAR-T cells were fresh or thawed at one year (365 days) after cryopreservation. (**B**) shows stimulation of BCMA.CAR-T cells by the BCMA positive tumor cell line U266 in the same setting. (**C**) shows the comparison of single positive IFNγ-, single positive TNFα- and double positive BCMA.CAR-T cells in the same setting. (Positive means BCMA.CAR-T cells which release certain cytokines, according to (**C**)).

**Figure 2 ijms-25-01394-f002:**
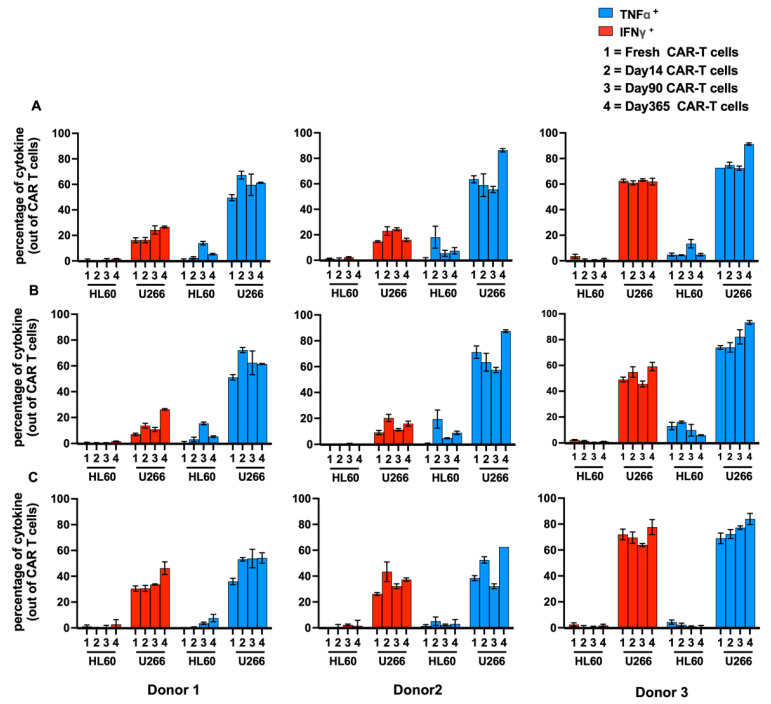
Cytokine-producing BCMA.CAR-T cells and different subsets of BCMA.CAR-T cells. After stimulation of different tumor cell lines, red and blue columns show the comparison of TNFα or IFNγ production in (**A**) all CD3^+^ BCMA.CAR-T cells, (**B**) CD4^+^ BCMA.CAR-T cells and (**C**) CD8^+^ BCMA.CAR-T cells at 4 timepoints (fresh or 14 days, 90 days and 365 days [one year] after cryopreservation) in each donor. Each column represents an individual donor.

**Figure 3 ijms-25-01394-f003:**
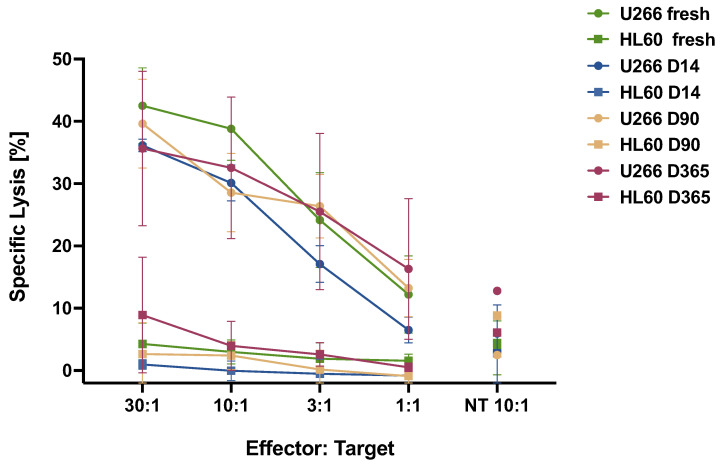
Chromium 51 release assay for the evaluation of cytotoxicity of BCMA.CAR-T cells. A BCMA^+^ U266 cell line or a BCMA^−^ HL60 cell line were incubated for 4 h. The curves depict the specific lysis of target cells by BCMA.CAR-T cells at different E/T ratios ranging from 30:1 to 1:1. There was no significant difference in lytic ability after 14/90 days or 1 year of cryopreservation (*F* = 3.385, *p* = 0.125). Non-transduced T cells (NT) served as a negative control at an E/T ratio of 10:1.

**Figure 4 ijms-25-01394-f004:**
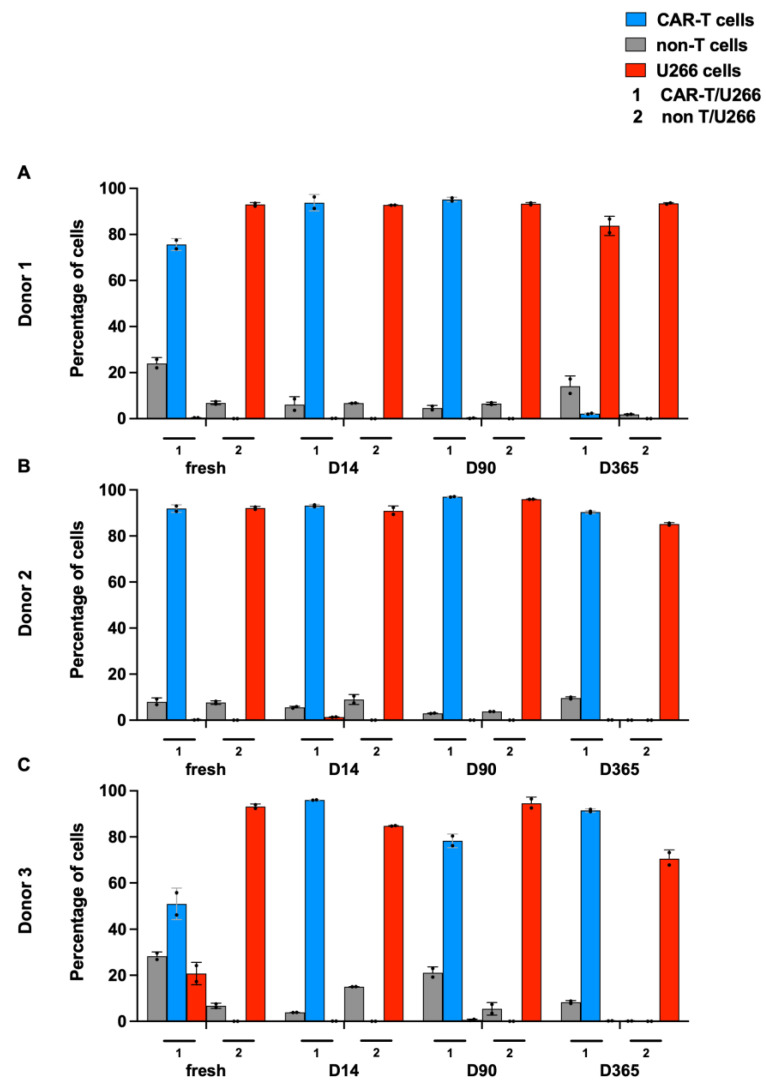
Cytotoxicity of BCMA.CAR-T cells as evaluated via co-culture assays. The y-axis indicates the percentage of BCMA.CAR-T cells versus non-transduced T cells as well as U266 cells at an E/T ratio of 1:4 and at day 10 (third round) of the process of co-culture. Two settings were tested: (1) CAR-T cell and U266 cells, with a certain percentage of non-transduced T cells, and (2) solely non-transduced T cells and U266 cells. (**A**–**C**) indicate the cytotoxicity of BCMA.CAR-T cells from different donors.

**Figure 5 ijms-25-01394-f005:**
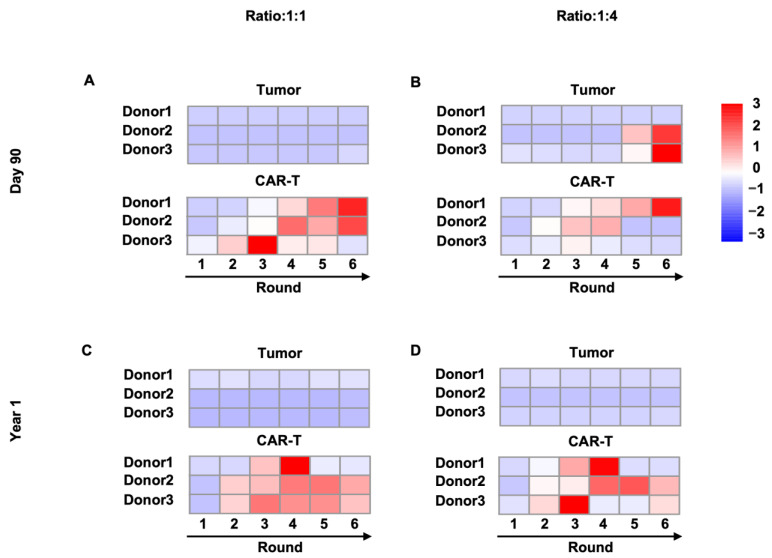
Heatmaps illustrate the absolute cell count (after normalization for each row) of BCMA.CAR-T cells and U266 cells during a full 6-round process of co-culture. The rounds of co-culture were performed with CAR-T cells thawed after 90 days or 1 year of cryopreservation. Colors from blue to red in the heatmaps show the change from less to more cells, thus indicating the trend of the proliferation versus lysis of cells. BCMA.CAR-T cells were thawed after 90 days and assays were performed at E/T ratio of (**A**) 1:1 or (**B**) 1:4 and BCMA.CAR-T cells thawed after 1 year were performed at E/T ratio of (**C**) 1:1 or (**D**) 1:4.

**Figure 6 ijms-25-01394-f006:**
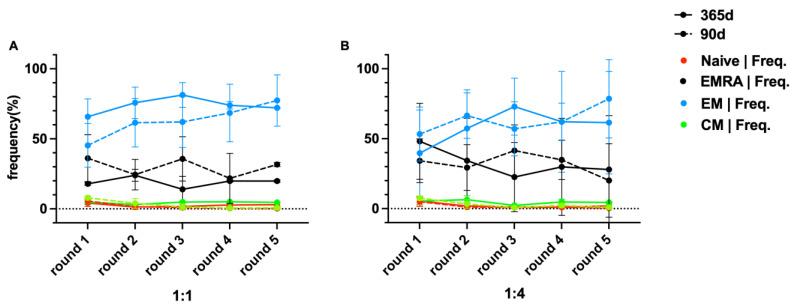
Dynamics in frequency of different T cell subsets during the first 5 rounds of co-culture. BCMA.CAR-T cells, which were thawed after different periods of time (90 days [broken lines] and 1 year [solid lines]), were divided into 4 subsets, naïve T (T_N_), central memory T (T_CM_), effector memory T(T_EM_) and effector memory RA^+^ T(T_EMRA_) cells. BCMA.CAR-T cells were co-cultured with U266 cells at different E/T ratios (**A**) 1:1 and (**B**) 1:4.

**Figure 7 ijms-25-01394-f007:**
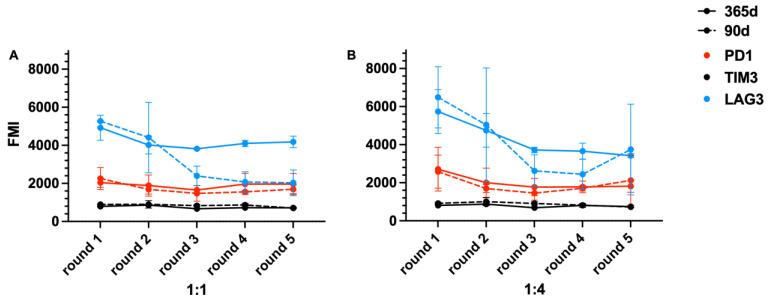
Trends in exhaustion markers on BCMA.CAR-T cells during the first 5 rounds of co-culture. BCMA.CAR-T cells were thawed after different periods of time (90 days [broken lines] and 1 year [solid lines], FMI [flow cytometry mean fluorescence intensity]). These surface markers illustrated the exhaustion or activation of BCMA.CAR-T cells during the co-culture. BCMA.CAR-T cells were co-cultured with U266 cells at different E/T ratios: (**A**) 1:1 and (**B**) 1:4.

**Table 1 ijms-25-01394-t001:** Transduction efficiency of BCMA.CAR-T cells at different timepoints. There was no significant difference in transduction efficiency after 14/90 days or 1 year of cryopreservation (*F* = 0.739, *p* = 0.49).

Transduction Efficiency of BCMA.CAR-T Cells	Fresh	D14	D90	D365
**Donor 1**	85.1	87.3	87.3	88.8
**Donor 2**	83.2	81.6	84.4	79.6
**Donor 3**	54.5	53.8	54.0	50.2

## Data Availability

Data contained within the article.

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
