# Peer review of "In Vitro Functionality and Endurance of GMP-Compliant Point-of-Care BCMA.CAR-T Cells at Different Timepoints of Cryopreservation"

_ijms, 2024, doi:10.3390/ijms25031394_

Round 1

Reviewer 1 Report

Comments and Suggestions for Authors

The authors have submitted a manuscript on BCMA.CAR-T cells. They present data on functionality and persistence at different time points of cryopreservation. For this purpose, BCMA.CAR-T cells were produced under GMP conditions using a CliniMACS Prodigy device and stored over nitrogen. 

Note 1: It is not clear from the presentation of the cells used here whether this is actually a point-of-CAR product or in-house production using a commercial product. 

It is intended to use these cells in a clinical trial (CARLOTTA). The cells must therefore be examined for their functionality, especially in connection with cryopreservation.

Remark 2: The work presented focuses strongly on cryopreservation and is purely in-vitro research. This must be made clear in the title of the paper.

The frequency of cytokine-producing cells was higher when stimulated with U266 cells (BCMA+).

Remark 3: Cytokine-producing cells were measured, not cytokine release, please name correctly.

Remark 4: The axes of Figures 1 A and B are not labeled.

IFN-gamma production of CD4+ CAR-T cells is lower compared to CD8+. 

Cytotoxicity was measured by 51Cr release assay, which unfortunately does not allow differentiation between CD4+ and CD8+ CAR-T cells. This is all the more remarkable given that coculture experiments were subsequently performed and cytometrically evaluated.

Remark 5: Are the results regarding cytotoxicity comparable?

This is followed by phenotyping data. These are not comprehensible on the basis of the methods provided.

Remark 6: Please provide the protocols for phenotyping in a comprehensible manner and with examples. For example, use the MIFlowCyt criteria as a guide. Please comment on the extent to which the stability of individual markers (e.g. CCR7) could influence the results.

Remark 7: There is a lack of data to determine transduction efficiency.

Author Response

Thank you very much for taking the time to review this manuscript. Please find the detailed responses in the attachment.

Reviewer 2 Report

Comments and Suggestions for Authors

Dear authors

The manuscript discusses the widely recognized B cell maturation antigen (BCMA)-targeted chimeric antigen receptor T (CAR-T) cell therapy for relapsed/refractory (R/R) MM, which has shown encouraging results. Despite the implementation of anti-BCMA CAR-T cell therapy, a considerable proportion of individuals diagnosed with multiple myeloma (MM) encounter a relapse. The complex characteristics of the tumor microenvironment, insufficient persistence of CAR-T cells, and antigen escape are primarily responsible for this.

Major comments

1. Nevertheless, the author emphasizes the long-term effectiveness of the treatment. Specifically, they discuss the follow-up period of 24 months for the combined infusion of anti-BCMA and anti-CD38 CAR-T cells, 268 days for the combined infusion of anti-BCMA and anti-CD19 CAR-T cells, and 42 months for the combined infusion of anti-BCMA and anti-CD19 CAR-T cells. The publication reports the administration of a mixture of anti-BCMA and anti-CD19 CAR-T cells for a duration of 248 to 966 days in Phase B. What is the main question being clarified by the completed investigation? This specific hypothesis should be included in the introduction.

2. In order to properly analyze and categorize the figures, it is necessary to divide them into distinct groups denoted by the labels A, B, C, and D. This division allows for a more systematic examination and comparison of the individual characteristics and features exhibited by each figure.

3.To guarantee conclusions that are in line with the evidence that is accessible, their professional knowledge and critical evaluation of the references being discussed.

Comments on the Quality of English Language

Extensive editing of English language required

Author Response

(The authors gave the same response as above.)

Round 2

Reviewer 1 Report

Comments and Suggestions for Authors

Thank you for modifying your manuscript.

Reviewer 2 Report

Comments and Suggestions for Authors

Though the authors do not understand the goal and are not satisfied with the response, the revised content is better than the original, even though this study sought answers to creativity compared to previous studies.

Comments on the Quality of English Language

 Minor editing of English language required